# Council of Europe Resolution on the Implementation of Pharmaceutical Care—A Step Forward in Enhancing the Appropriate Use of Medicines and Patient-Centred Care

**DOI:** 10.3390/healthcare12020232

**Published:** 2024-01-17

**Authors:** Martin C. Henman, Silvia Ravera, Francois-Xavier Lery

**Affiliations:** 1School of Pharmacy and Pharmaceutical Sciences, Trinity College, D02 PN40 Dublin, Ireland; 2European Directorate for the Quality of Medicines and HealthCare (EDQM), Council of Europe, F-67081 Strasbourg, France; silvia.ravera@edqm.eu (S.R.); francois-xavier.lery@edqm.eu (F.-X.L.)

**Keywords:** evidence-based pharmacy practice, health policy, pharmaceutical services, medication errors, community pharmacy, hospital pharmacy services

## Abstract

Pharmaceutical care was proposed to address morbidity and mortality associated with medicine-related problems. It utilises the pharmacist’s expertise in medicines, their relationship with the patient and cooperation with other healthcare professionals to optimise the use of medicines. The European Directorate for the Quality of Medicines & HealthCare (EDQM), part of the Council of Europe, found significant variation in the acceptance of pharmaceutical care and in the implementation of pharmaceutical care in Europe. A multidisciplinary group was established to draft a statement of principles and recommendations concerning pharmaceutical care. Through face-to-face meetings, circulation of draft texts and informal consultation with stakeholders, the group produced a resolution. On 11 March 2020, the resolution was adopted by the Committee of Ministers of the Council of Europe. It explains pharmaceutical care and illustrates pharmacists’ contribution to medicine optimisation in different care settings. Pharmaceutical care’s value to health services and its place in health policy were emphasised by addressing the risks and harms from suboptimal use of medicines. Pharmaceutical care can improve medicine use, promote rational use of healthcare resources and reduce inequalities in healthcare by realigning the roles and responsibilities of pharmacists and healthcare professionals. EDQM will promote and advocate for the implementation of pharmaceutical care by enacting practice Resolution CM/Res(2020)3.

## 1. Introduction

### 1.1. Pharmaceutical Care

Medicines are the most frequent intervention in healthcare systems around the world. They are used for treatment, symptom control and prevention and provide substantial benefits but are not without adverse effects and risk. Medicines are authorised for dispensing by prescription in primary care, in community pharmacies and in hospitals, or without prescription from community pharmacies. Selected medicines are available from non-pharmacy outlets in some countries. Pharmacists contribute to the safe use of medicines by patients and professionals through their evaluation of the quality of medicines and of the appropriateness of prescriptions during dispensing and through their engagement with people considering using medicines for self-care [1,2,3,4].

Most problems with the use of medicines stem from the way they are selected and used by the prescriber, the pharmacist and the patient [5]. Pharmacists are often seen simply as providers of medicines with limited authority to intervene to improve the use of medicines [1,3,4]. These factors may be among the reasons for health systems not sufficiently mitigating the risks and harms that stem from irrational use of medicines or polypharmacy. Recognising this, the World Health Organization [WHO] selected Medication Without Harm as its third global patient safety challenge [6].

Pharmaceutical care was proposed in 1990 by Hepler and Strand as ‘the responsible provision of drug therapy for the purpose of achieving definite outcomes that improve a patient’s quality of life’ in response to the increase in morbidity and mortality associated with medicine-related problems [7]. It utilises the pharmacist’s knowledge of medicines, their relationship with the patient and cooperation with other healthcare professionals to address the outcomes of medication use, including their impact on the patient’s quality of life. Pharmacists routinely providing pharmaceutical care to individuals could contribute significantly to the health of the population.

### 1.2. Development of the Resolution CM/Res(2020)3

The Council of Europe was founded in 1949 to promote and advocate for human rights, democracy and the rule of law. In 2007, the Council of Europe Committee of Ministers agreed to transfer its activities in the field of Social and Public Health to the European Directorate for the Quality of Medicines (EDQM). Subsequently (2009), the EDQM modified its name to the European Directorate for the Quality of Medicines and HealthCare and established the European Committee on Pharmaceuticals and Pharmaceutical Care (CD-P-PH) (steering committee) [8].

One of its subordinate bodies, the Committee of Experts on Quality and Safety Standards in Pharmaceutical Practices and Care (CD-P-PH/PC), found that there was significant variation in the acceptance of pharmaceutical care among stakeholders outside of pharmacy organisations and in the implementation of pharmaceutical care in Europe [9,10,11]. Consequently, it established a working group in 2018 to draft a statement of principles and recommendations concerning pharmaceutical care in a Council of Europe legal instrument known as a resolution. This multidisciplinary working group consisted of hospital and community pharmacists, academics and representatives of national competent authorities from 15 different countries. These experts drew on their knowledge of the literature, supplemented with searches for updated evidence, their familiarity with the policies and procedures of their countries and their expertise to outline the structure and content of a draft resolution. Through a combination of face-to-face meetings and circulation and revision of draft texts within the working group and to the CD-P-PH/PC, and through an informal consultation process with stakeholders successive drafts were amended. The working group agreed on a final draft resolution. This was accepted by the CD-P-PH/PC and approved by its steering committee, the CD-P-PH. Following a legal and structural review and translation into French, the resolution was submitted to the Committee of Ministers of the Council of Europe and adopted by on 11 March 2020 [12]. The resolution is provided in Appendix A

This paper aims to describe the resolution and to illustrate the concept of pharmaceutical care, its relevance to health services and potential contribution to health policy.

## 2. Discussion

### 2.1. Pharmaceutical Care and Patient Care

Pharmacists have a duty of care to ensure that any medicine—prescription or non-prescription—they provide to a patient is appropriate for them. Medicines may be used for treatment but also for prevention. The patient’s appreciation of their symptoms or conditions and of the role that medications play are pivotal in determining when and how they decide to take their medications [13]. The autonomy of the patient using medicines is often overlooked. Patients must agree to take them, and they expect to be told the reason, the benefits and side effects they may experience and what they should do about them [13]. In pharmaceutical care, the pharmacist collaborates with the patient to develop, agree, implement and monitor a therapeutic plan. The pharmacist communicates and collaborates with other healthcare professionals involved in medicine use to coordinate the implementation of this therapeutic plan. The pharmacist takes responsibility for addressing the actual and potential medicine needs of the patient and also for attaining the best outcomes and linking these to improving the patient’s quality of life. The therapeutic relationship at the heart of pharmaceutical care goes beyond provision and enables the patient to entrust the pharmacist with their medicine needs and, in return, to make use of the pharmacist’s expertise and their advocacy. Pharmaceutical care is therefore a patient-centred process based upon a therapeutic relationship with the patient and a collaborative one with other healthcare professionals [9].

Prescriptions are requests, not orders, to dispense, but they sometimes do not include the intended use of the medicine or other information important for checking the dose, such as the age or weight of the patient [14]. Nevertheless, pharmacists must screen prescriptions for clinical, legal and administrative problems and begin by talking with the patient. This screening process is a key safety measure for which pharmacists are responsible and for which they are held accountable [4,15]. To do this effectively, relevant information must be shared between prescribers and pharmacists, so that they can collaborate and provide quality care in every setting in which medicines are used. Many prescription medicine problems can be easily resolved by giving pharmacists the authority, within an agreed framework, to adapt a prescription, to change the dose, formulation, or regimen, to renew or extend prescriptions in order to maintain the continuity of care and to make a therapeutic substitution of a medicine with one from the same therapeutic class [16]. By resolving and preventing medicine-related problems, pharmacists enhance medication safety and increase the efficiency of medicine provision, as demonstrated during the COVID-19 pandemic [17,18]. However, there remains considerable variation across Europe in the scope of practice of pharmacists which prevents them from contributing fully to patient care and health services from gaining these benefits [19,20].

Many patients receive care with medicines from several healthcare professionals and may also take non-prescription medications. Gathering information about these medications and identifying the patient’s understanding of their need for medicine falls to the pharmacist. The use of multiple medicines (polypharmacy) poses increasing risks [21] and needs to be reviewed regularly [22], and consequently deprescribing may be necessary [23]. Patients frequently need advice, support and advocacy to be able to manage their medicines safely and effectively in their daily routine. Medication adherence is on average around 50%, and in the long term adherence declines further. Research shows that adherence is dependent upon discussion of the issues that the medicine user thinks important [13]. When patients move between healthcare settings, the risk of medicine problems increases and must be addressed [24]. Pharmaceutical care applied at crucial moments in the provision of care is a quality-enhancing process; it will support everyone involved in a patient’s medication use; it will improve the coordination of care; and it will benefit the health service by maximising the pharmacist’s contribution to optimising medicine use in all healthcare settings.

### 2.2. Pharmaceutical Care Services in the Community Pharmacy Setting

Community pharmacists are usually independent contractors operating in a retail setting, and consequently they are often not recognised as providers of health services and are omitted from health service consideration in both policy and practice [16]. In reality, however, community pharmacists are a first point of contact with the health service and see patients more often than other healthcare professionals [16]. They assess, screen, treat, refer and signpost patients with symptoms, concerns and medicine needs. They sustain primary care by ensuring the quality of medicines, reviewing prescriptions and providing advice to prescribers and by the responsible provision of prescription and non-prescription medicines and advice. They undertake screening for disease risk factors and provide monitoring for prevalent conditions through the use of validated tools and measurements [20,25]. Their accessibility and support for self-care is used to enable assessment and treatment of minor illnesses and, if necessary, to refer patients on to more specialist assessment [26,27]. Pharmaceutical care is a holistic approach and so pharmacists advise on and recommend healthy behaviours and lifestyle changes alongside medicine advice and counselling. These two classes of service expand the capacity of primary care and establish effective and efficient pathways of care for people at an appropriate time and place.

In chronic and complex cases, an in-depth review of medicines together with timely communication are essential to the coordination and continuity of care [28]. Specific services, such as advice and help to improve the use of medical devices such as respiratory inhalers [29] or the management of chronic diseases such as diabetes mellitus or asthma through focused programmes [16] ensure that treatment problems are resolved, adherence improved and treatment intensification delivered. The same goes for the pharmaceutical care service for patients to whom innovative new medicines have been prescribed [30,31]. Patients’ conditions and medication needs change over time so community pharmacists provide ongoing support and monitoring. Their referral of patients for further assessment and treatment facilitates the appropriate provision of care at the earliest opportunity and improves the quality of care. Outpatients live at home but attend hospital clinics and their medicine needs, such as oral chemotherapy, can be complex [32,33], requiring community pharmacists to liaise with hospital pharmacists and prescribers to resolve problems and coordinate medicine use.

Collaboration between health services and community pharmacists has delivered harm reduction programmes of consistent quality in Europe [20,25], such as opioid substitution [34], needle exchange [35] and smoking cessation [36]. Similarly, experience with vaccination services has shown that they increase the uptake of vaccines in the population and are valued by the public [37,38]. Their success has led more countries to enable pharmacists to vaccinate during the COVID-19 pandemic [20]. These complex services have delivered improvements in public health when supported by accredited training and qualifications, promotion and information technology.

### 2.3. Pharmaceutical Care Services in the Hospital Pharmacy Setting

Pharmacists in hospitals are responsible for the procurement, distribution and compounding of medicines and, in their roles as clinical pharmacists, to optimise the use of medicines in care through individual and institution-wide interventions [39]. Pharmacists are key contributors to the Pharmacy (or Drug) and Therapeutic Committees in hospitals that develop evidence-based policies and protocols about medicine use that are collated in guides and provide medication information services [40,41].

On admission to hospital, a patient’s current medicine use and history must be assessed before changes can be made to their medicines. Pharmacists conduct medication reconciliation on admission as part of the medication review carried out by the multidisciplinary team involved [42]. The needs of patients who remain in hospital change and pharmacists, working alongside the medical and nursing staff, provide further reviews and recommendations concerning the choice of medication, dose, route of administration and monitoring [43]. Pharmacists can resolve many medication problems speedily and efficiently when they have authority to modify prescriptions within the hospital’s agreed guidelines.

Medicine use in hospitals is substantial not only in volume and cost, but also in potential risk and harm [44]. Hospitals care for patients with multiple morbidities, with advanced disease, those requiring a transplant, those with rare conditions and also those with urgent needs [45]. Each of these circumstances may be complicated by poor organ function (e.g., liver or kidneys), frailty and other vulnerabilities [46]. The diverse specialities, advanced practice and innovative treatments that characterise hospital care have an impact on the medicines used. The range of possible permutations of these factors contributes to the scale of the complexity of the medicine needs that must be assessed and met over the course of the patient’s stay. The multidisciplinary teams must include a pharmacist to help resolve and mitigate the actual and potential problems that arise from the competing and conflicting clinical demands. The establishment of advanced practitioner qualifications and roles for pharmacists [47,48] and programmes of intensive and comprehensive pharmaceutical care coordinated and delivered by hospital pharmacists have been shown to produce benefits that include reduced morbidity and length of stay with a significant return on the investment made [49,50].

In many countries, pharmacists contribute to the provision of palliative care in hospices and hospitals [51].

Discharge to home or to a residential care facility requires counselling of the patient and coordination with other healthcare providers if these transitions are to be safely completed and unplanned readmissions are to be avoided [52].

The demand for residential care will continue to increase and the staffing of these facilities is often dependent upon a few in-house professionals and a number of external care providers. The regular involvement of pharmacists in the review, selection and monitoring of medicines is essential given the prevalence of polypharmacy, multiple morbidity, frailty and impaired cognitive function that has been shown to result in medicine-related problems [53].

### 2.4. Pharmaceutical Care and the Health Service

Throughout Europe, health systems are under pressure from increased demands, healthcare cost inflation, adverse medical events and global threats to public health. Quality health services are those that are people-centred, equitable, effective, integrated, safe, timely and efficient [54]. Quality medications and quality medication use processes are key components of quality health services [55]. Although the value of quality medications is recognised by health services, across Europe, the pattern of medication use is consistently poor and constitutes a burden on health services [56].

Pharmacists protect and maintain health services through their stewardship of medicines, including antimicrobial stewardship [57], the provision of medicine information and advice about treatment procedures tailored to support the appropriate use of medicines by patients and by professionals [19], the operation of pharmacovigilance systems for detecting and responding to the adverse effects of medicines [58] and their work to counter threats from medicine shortages [59] and falsified medicines [60]. Pharmacists also provide patient services in Europe, but few of these are supported by health services which limits their sustainability and their impact [20]. The pharmacists’ scope of practice is comprehensively regulated throughout Europe [19] but frequently pharmacists’ roles in patient care have not been incorporated into these regulations or accepted by health services. Consequently, pharmaceutical care has been ignored as an agent of change in the majority of European countries.

At present, responsibility and authority for the medicine use process are poorly defined, and are dispersed between the patient and different healthcare professionals involved along the care pathway so that the quality of the process and its outcomes are easily compromised. The integration of pharmaceutical care into health service policies will ensure that the pharmacist’s expertise and skills are utilised to optimise individual patient care and to improve the safe and efficient use of medicines throughout the health service. However, because pharmaceutical care is a collaborative process, pharmacists cannot implement and sustain it in all health service settings without cooperation and support at every level of the health service. Examples of the impact of pharmaceutical care are presented in Table 1.

The pace, scope and scale of change in health services in Europe have been insufficient to deliver accessible, equitable and high-quality services that protect patient safety in a sustainable manner [61]. Delivering the necessary changes in each country will require a combination of methods tailored to its particular needs [54,61]. Improving health services requires the utilisation of all of the expertise available and the cooperation of all of the actors involved. Therefore, public and private providers of health services, governments, regulatory bodies and healthcare professional associations all have a part to play and must contribute to the implementation of pharmaceutical care.

Implementation must be managed to translate aims into outcomes. For pharmaceutical care, each of the following factors must be addressed:Integration of pharmacists and pharmaceutical care within the health services. Pharmacists, as experts in medicines, already advise and inform patients about the use of medicines, but are underutilised in service development and delivery [19,62]. Recognition of pharmacists as providers of care and as members of multidisciplinary teams is essential to maximising use of their skills in a patient-centred health service [63]. Community and hospital pharmacists, as well as those in primary care practices, residential care homes and detention settings, should have the authority to deliver pharmaceutical care and a framework within which to do so.Evaluation of the scale of poor quality of medication use (prescription and non-prescription) and its impact on the health service using a systems approach [54]. Inadequate data and data in siloes contributes to the lack of comprehensive and holistic assessment of these data with consequently inadequate responses to poor medicine use [55,64]. Policies should include pharmaceutical care and leverage pharmacist expertise both nationally and locally to optimise medicine use and address medicine shortages and public health emergencies.Interprofessional collaboration needs to be incorporated into practice in health services. It must be mandated and supported in co-located, but especially in dispersed, practices. Interprofessional collaboration is integral to pharmaceutical care and helps ensure the coordination and continuity of service provision [62,65,66].Digital systems and procedures to connect pharmacists should be an integral part of the implementation of eHealth policies. This will remove barriers that prevent the sharing of data needed for safe and effective care. Support for both face-to-face and remote consultations and recording of pharmaceutical care interventions will capture data essential for the monitoring and evaluation of service quality.Realigning services and resources to support the role of pharmacists as care providers. For example, community pharmacists are only considered part of primary care in a few countries and are remunerated in most of them, mainly for the dispensing of medicines. Hospital pharmacy departments have their staffing complements and budgets based on the purchase and distribution of medicines and not on the need for clinical pharmacy services. The focus on the supply of medicines is reinforced by misaligned resourcing which creates inefficiencies and is a major barrier to the introduction of pharmaceutical care and to change throughout health services [67].Workforce planning to ensure an adequate supply of pharmacists and other pharmacy team members. Pharmacy is often excluded from consideration because it is seen as a support service by other healthcare professionals and health service managers. This leads to gaps in the skilled workforce necessary for the implementation of pharmaceutical care, the optimisation of medicine use and the development of the health service [68].Continuous professional development (CPD) for pharmacists to support the change in their scope of practice, improve collaboration and provide high-quality, standardised and appropriately certified healthcare services, e.g., vaccination.Commitment from health services to establish and sustain the facilities in which the clinical experiential placements and interprofessional education necessary for pharmaceutical care can be delivered. Pharmacy education uses competency frameworks to prepare pharmacists to deliver pharmaceutical care and advanced practitioners but needs support from the health services, professional bodies and academic institutions to create the foundations for cooperation and collaboration in professional practice [68,69].Pharmaceutical care programme development and continuous quality improvement (CQI). Evaluation and CQI measures should be used to monitor implementation, to quality-assure service delivery and to assess the outcomes.

**Table 1 healthcare-12-00232-t001:** Examples of the impact of pharmaceutical care.

Concept, Sector.	Problem(s)	Intervention(s)	Impact(s)
**Pharmaceutical Care**	The medicine use process from medicine selection, prescribing and dispensing to patient education is a series of uncoordinated steps to supply the medicine. This leads to suboptimal medicine selection, use and monitoring. At present, policy initiatives tend to focus on a single step which produces limited benefits and unintended consequences.	Pharmacist offers to take responsibility for the quality of medicine use and for the health outcomes. Pharmacists have the necessary expertise and share the duty of care with the prescriber for prescription medicines and with the patient for non-prescription medicines.	Creates a coordinated, structured process that resolves problems with medicines through increasing patient-centred care provided by pharmacists and enhancing collaboration with prescribers [10,15].
**Primary Care**	Fragmented primary care systems lead to ineffective episodes of care, medication errors and unnecessary referral to hospital. Shortages of prescribers in primary care lead to poor access and excessive workload.	Pharmacists working in their community practices can provide access to preliminary assessment, screening, treatment of minor illnesses, medication review, monitoring, support and appropriate referral.	Care can be provided when it is needed, at the level that is needed [18,65]. Those with chronic conditions can be cared for at home for longer and vulnerable groups can be cared for in the community [16,31]. More complex cases can be referred to the appropriate service and unnecessary admissions avoided [16,64].
**Acute Hospital Sector**	Patients with multiple morbidity, those with acute conditions and those requiring intensive monitoring all need medicines—these may be combinations of medicines, specialist medicines for acute and high-risk conditions and medicines that must be monitored intensively.	Medicine reconciliation on admission, integrating clinical pharmacists with specialist teams, reviewing and monitoring medicine use and outcomes in vulnerable patients, medicine review and counselling at discharge [42,43].	Reduction in medicine-related errors and problems [5]. Improved patient safety at transitions of care and in high-risk circumstances [24]. Optimisation of medicine use increases the quality of care and reduces the length of stay and unscheduled readmission [49,50].
**Residential Care**	Patients unable to live independently often receive multiple medicines (polypharmacy) but may be infrequently assessed. Unnecessary and inappropriate medicines can reduce the patient’s quality of life and increase the likelihood of hospital admission.	Medication review by pharmacists and multidisciplinary team reviews to optimise medicines. Guidance and advice for care teams concerning medicine use, monitoring and discontinuation.	Reduction in polypharmacy and inappropriate medicines will decrease prescribing cascades. Improved medicine use policies will reduce the burden of medicine management on care staff [52,53].
**Public Health**	Patients with minor illnesses attending general practitioners unnecessarily add to workload and consume scarce resources. Patients with symptoms not assessed and appropriately referred in a timely manner.	Programmes to assess and treat minor illnesses based in community pharmacies. Screening, monitoring and referral, for example, for blood pressure and atrial fibrillation.	Reduced costs and effective treatment of minor illnesses [27]. More effective use of pharmacists and general practitioners [20]. Early detection, reduction in uncontrolled conditions and appropriate referral [25].
	Response times to epidemics and access to vaccinations have been shown to be inadequate.	Vaccine administration through community pharmacies.	Increased proportion of population vaccinated [20,37].
	Poor access to medicines that are time-critical, such as adrenaline for serous allergic reactions and emergency hormonal contraception.	Medicines for emergency administration made available through community pharmacies.	Increased access and availability of medicines for emergency situations.

### 2.5. Pharmaceutical Care and Health Policy

Pharmaceutical care is a statement of principle, a change in the scope of practice that mandates pharmacists to provide patient-centred care and to optimise medicine use through structured patient engagement and interprofessional collaboration. It will enable them to make a greater contribution through their interventions and those of the teams in which they are included. The scale of poor quality of medicine use is underestimated, but its effects, reduced therapeutic benefits, increased adverse effects and abuse of medicines are all adding to the rising prevalence and cost of medicine-related harm. Much of this harm is preventable, and the consequent expenditure avoidable, if a culture of responsible use of medicines is prioritised and promoted. Universal Healthcare, strengthened primary care, integrated care, interprofessional collaboration and comprehensive patient safety programmes are all policies to which pharmaceutical care can contribute [69]. However, systemic and entrenched barriers to pharmaceutical care persist, including regarding the roles and responsibilities of the different healthcare professionals. For pharmacists, this regards their scope of practice, their remuneration, their access to relevant patient and clinical information, their integration in the care pathway and suitable planning of their workforce. The legacy of health service structures, institutions and regulations underpinned by traditional practices, accepted norms and established roles and responsibilities supports the status quo and impedes health service reform. Health services cannot resolve all of these problems and need political leadership to drive the necessary regulatory changes, provide strategic direction and initiate health service reform, particularly for the following aspects:Vision of pharmacists as providers of pharmaceutical care. Pharmaceutical care is a patient-centred, collaborative model of care that harnesses the pharmacist’s expertise to tackle the pervasive problem of poor medicine use—the societal consequences of which are growing. Pharmaceutical care is a key policy to address this problemChanging pharmacists’ scope of practice. New models of care, new technology and new medicines challenge traditional forms of professional practice throughout healthcare. Not only may legal and regulatory change be required but also substantial cultural and organisational change. Policy makers’ engagement is essential to establishing constructive engagement from stakeholders and achieving practical reform of scopes of practice to deliver patient-centred health services.Health financing and resource allocation should promote patient care. Many aspects of the funding for healthcare systems discourage and impede pharmacists from providing pharmaceutical care. Instead, they focus on medicine availability and costs. Pharmaceutical care and reform of medicine supply schemes should go hand-in-hand in order to optimise the use of resources. This major change can only be delivered with the support of all the relevant government departments.Integration of pharmaceutical care in every setting in which medicines are used. Fragmentation and uncoordinated healthcare delivery exacerbate poor medicine use and reduce the effectiveness and efficiency of health services. Pharmacists, as medicine experts and providers of pharmaceutical care, should be utilised throughout government and health services to develop policy and to oversee its implementation. eHealth policies must enable pharmacists to access, share and contribute to the exchange of health information upon which integrated care depends.Comprehensive workforce planning and education reform to provide enough pharmacists and advanced practitioners to meet the needs of the health services, public health policy, academic practice and the pharmaceutical industry [68,69]. Pharmacy is a relatively small profession (compared to medicine and nursing) and while many countries recognise the need for a skilled workforce for the pharmaceutical industry, few are making adequate provision for the pharmaceutical needs of health services or have a vision for research and development that will also stimulate innovation in patient care.

## 3. Further Work with the Resolution

A resolution is a type of ‘soft law’ used by intergovernmental bodies to bring together countries to tackle important, common problems. It is not legally binding. Instead, by setting out agreed principles and statements, it represents a commitment to develop and enact policies to address the subject of the resolution. EDQM has worked to promote and engage Member States within the Council of Europe concerning the resolution.

Two webinars have been held; the first focused on the resolution and its content [70] and the second on the resolution and examples of pharmaceutical care from community and hospital practice [71].

EDQM has collaborated with members of the South Eastern European Health Network—Albania, Bosnia and Herzegovina, Bulgaria, Moldova, Montenegro, North Macedonia, Romania and Serbia [72]. They carried out a survey to assess the implementation of pharmaceutical care in daily practice as described in the resolution. The awareness and implementation of pharmaceutical care and related services varied significantly across the different countries. Among the most frequently reported barriers were the absence of regulations or healthcare policies, the lack of access to patient medical records, the limited data on the benefit of pharmaceutical care and the lack of infrastructure such as computerised systems or IT tools.

EDQM, through the Committee of Experts on Quality and Safety Standards in Pharmaceutical Practices and Care (CD-P-PH/PC), is actively working on guidance concerning medication review which is an essential part of pharmaceutical care. It is also examining remote and online access to medicines, an emerging and challenging area for policy makers.

## 4. Conclusions

Pharmaceutical care is a policy that can improve the use of medicines, promote the rational use of healthcare resources and help reduce inequalities in healthcare. It does so by recognising the roles and responsibilities of all healthcare professionals (in particular of pharmacists) in the responsible use of medicines. EDQM considers that this will be possible by putting into practice Resolution CM/Res(2020)3 on the implementation of pharmaceutical care. Progress was impeded by the COVID pandemic and given the complexity and entrenched barriers to health policy reform, it will require the combined intervention of governments and health services in collaboration with the relevant associations of healthcare professionals. As an intergovernmental body with a leading role in standard setting and guidance concerning the quality of medicines and healthcare, the EDQM will continue to promote and advocate for the implementation of pharmaceutical care and is ready to cooperate with governments, national competent authorities, partners and healthcare stakeholders throughout Europe.

## Data Availability

Data are contained within the article and Appendix A.

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
