# Peer review of "Council of Europe Resolution on the Implementation of Pharmaceutical Care—A Step Forward in Enhancing the Appropriate Use of Medicines and Patient-Centred Care"

_healthcare, 2024, doi:10.3390/healthcare12020232_

Round 1
Reviewer 1 Report
Comments and Suggestions for Authors
The article entitled “Council of Europe Resolution on the implementation of pharmaceutical care – a step forward in enhancing the appropriate use of medicines and patient-centred care describe the resolution and illustrate the concept of pharmaceutical care, its relevance to health services and potential contribution to health policy. The article is clear and, in my opinion, focus different points to considered for the activity of the pharmacist, highlighting the importance of them for the patients. I would like, only, to suggest a modification to improve the abstract. So, in line 23 the authors should spell out the abbreviation EDQM, In line 13 the authors have written European Directorate for the quality of medicines & health care or the authors define here or write in full in the line 23, for reader understand the text.
Comments on the Quality of English LanguageMinor editing of English language required
Author Response
|
Reviewer 1 |
|
|
|
|
|
Section |
Line |
Reviewer comment |
Response |
Amendment -line number(s) |
|
Abstract |
13 |
the authors have written European Directorate for the quality of medicines & health care or the authors define here or write in full in the line 23 |
The abbreviation has been added to line 13. |
Line 13 |
|
|
23 |
in line 23 the authors should spell out the abbreviation EDQM, |
As above. |
See above. |
Reviewer 2 Report
Comments and Suggestions for Authors
Socially relevant and well-justified topic, but it lacks a better theoretical framework and explanation of the methodology used to design the resolution and to illustrate the concept of pharmaceutical care.
The methodology must be more explicit regarding the sociodemographic and professional characteristics of the participants, details of the type of documents used in the analysis, identification, screening and selection process, eligibility criteria, type of document analysed and consensus criteria on the proposals presented. This information is useful for an interpretative analysis in the discussion.
The conclusion should give a clearer answer to the objective of the article.
In the list of references, citations 70 and 71 are not cited in the text and 72 does not exist in the list of references. Some of the links to the reference lists are not available.
Author Response
|
Reviewer 2 |
Reviewer comment |
Response |
Amendment -line number(s) |
|
|
Introduction |
|
The methodology must be more explicit regarding the sociodemographic and professional characteristics of the participants, details of the type of documents used in the analysis, identification, screening and selection process, eligibility criteria, type of document analysed and consensus criteria on the proposals presented. |
We thank the reviewer for this interesting comment. The creation of a Resolution (a type of soft law) is not a process of research but one of discussion, consensus, consultation and finalisation. We have added more detail to the method and have added more explanation of the nature of a Resolution. |
Introduction: Lines 71-77.
Discussion: Lines 366-386. |
|
Conclusion |
|
The conclusion should give a clearer answer to the objective of the article. |
We have reformulated part of the conclusion. |
Lines 391-393. |
|
Other comments |
|
citations 70 and 71 are not cited in the text |
They have been cited. |
|
|
|
72 does not exist in the list of references |
This has been addressed. |
|
|
|
|
Some of the links to the reference lists are not available |
We have checked these and all of the links are correct and functioning.
|
|
|
Reviewer 3 Report
Comments and Suggestions for Authors
Thank you for the opportunity to review this promising manuscript, which contains significant content that I feel will be of high interest to readers after some improvements.
First, the introduction could better clarify this article's rationale or gap addressed (e.g., increasing awareness of CoE resolution, and translating the resolution to practical implementations). I think this may also help the authors identify what information from the introduction subsections might be peripheral to make their points.
Second, the resolution was passed 3+ years ago and therefore should attempt to highlight the impact the resolution already has had rather than just what its potential impact is. I realize the COVID pandemic put a lot of things on pause, so perhaps this again ties into the rationale for this piece.
Third, please consider ways to improve the readability of this article. You might also consider a table or figure crossing each pharmaceutical care section with each component of your intended objective (e.g., illustrative example, relevance to health services, contributions to health policy). There are also several typos and opportunities to be more concise/direct in language throughout the paper.
Comments on the Quality of English LanguageThere are several typos and opportunities to be more concise/direct in language throughout the paper.
Author Response
|
Reviewer 3 |
Reviewer comment |
Response |
Amendment -line number(s) |
|
|
Introduction |
|
The introduction could better clarify this article's rationale or gap addressed (e.g., increasing awareness of CoE resolution, and translating the resolution to practical implementations) |
We thank the reviewer for this comment but would point out that it is addressed – “…. that there was significant variation in the acceptance of pharmaceutical care among stakeholders outside of pharmacy organisations and in the implementation of pharmaceutical care in Europe [9-11].” We have amended the final sentence: “This paper aims to describe the resolution and to illustrate the concept of pharmaceutical care, its relevance to health services and potential contribution to health policy.”
|
Lines 64-67.
Lines 82-83. |
|
Section 3 |
|
‘the resolution was passed 3+ years ago and therefore should attempt to highlight the impact the resolution already has had rather than just what its potential impact is. I realize the COVID pandemic put a lot of things on pause, so perhaps this again ties into the rationale for this piece.’ |
We thank the reviewer for this perceptive comment. Indeed the process of bringing this paper to the point of submission was delayed and the implementation of many health policy reforms were substantially impacted by the COVID 19 pandemic. It would, however, be hard to ascribe policy changes made to the resolution, rather, EDQM works continually with governments and stakeholders to promote its implementation. We have added some explanation about the nature of a resolution and detail of the activities taken to raise awareness of the Resolution. |
Lines 366-386. |
|
Section 2.4 |
|
‘consider a table or figure crossing each pharmaceutical care section with each component of your intended objective (e.g., illustrative example, relevance to health services, contributions to health policy)’ |
Thank you for this comment. We have added a table to provide illustrative examples as suggested. |
Table 1. |
|
|
|
There are also several typos and opportunities to be more concise/direct in language throughout the paper |
Thank you, we have searched the manuscript and addressed these. |
|
Round 2
Reviewer 3 Report
Comments and Suggestions for Authors
Several comments were not addressed by the authors in the revised submission:
1. Lack of reference for a strong statement in lines 41-42. Soften language if you don't have a citation.
2. Unclear what the 'it' at the beginning of the sentence on line 42 refers to...lack of recognition as providers or limited authority or both?
3. Lns 47-54: It has been 30+ years since Helpler and Strand and pharmaceutical care have evolved (some may say co-opted). Why are you not citing more contemporary resources and updated definitions?
4. Ln 220: Why is people-centred more appropriate than patient-centered here, given you only use the former once and the latter six times throughout the paper?
5. Ln 358: Are you comparing pharmacy to other healthcare professions when stating the profession is relatively small?
Author Response
Responses to Reviewer 3
- Lack of reference for a strong statement in lines 41-42. Soften language if you don't have a citation.
The sentence has been amended.
Pharmacists are often seen simply as providers of medicines with limited authority to intervene to improve the use of medicines [1,3-4].
- Unclear what the 'it' at the beginning of the sentence on line 42 refers to...lack of recognition as providers or limited authority or both?
It refers to both and has been amended.
These factors may be among the reasons for the health systems not mitigating enough the risks and harms that stem from irrational use of medicines or polypharmacy.
- Lns 47-54: It has been 30+ years since Helpler and Strand and pharmaceutical care have evolved (some may say co-opted). Why are you not citing more contemporary resources and updated definitions?
EDQM assumed responsibility for healthcare in 2007 and the Hepler and Strand formulation was the accepted definition of pharmaceutical care at that time. EDQM took this definition as the guiding statement for the responsible committee, the European Committee on Pharmaceuticals and Pharmaceutical Care (CD-P-PH). As your comment states some of the more recent definitions can be considered to have been co-opted. EDQM is concerned with the needs citizens and societies and is not aligned to any group and continues to use the Hepler and Strand definition.
- Ln 220: Why is people-centred more appropriate than patient-centered here, given you only use the former once and the latter six times throughout the paper?
When health systems are referred to as a whole, the term people-centred is used. It is used in the cited reference from the Institute of Medicine and by other organisations such as the WHO.
- Ln 358: Are you comparing pharmacy to other healthcare professions when stating the profession is relatively small?
Yes. The sentence has been amended.
Pharmacy is a relatively small profession (compared to medicine and nursing) and while many countries recognise the need for a skilled workforce for the pharmaceutical industry, few are making adequate provision for the pharmaceutical needs of health services or have a vision for research and development that will also stimulate innovation in patient care.